# Pharmacological Prevention in Breast Cancer: Current Evidence, Challenges, and Future Directions

**DOI:** 10.3390/cancers17223597

**Published:** 2025-11-07

**Authors:** Samanta Sarti, Alessandro Adriano Viansone, Olga Serra, Chiara Casadei, Lorenzo Cecconetto, Giandomenico Di Menna, Alberto Farolfi, Caterina Gianni, Marita Mariotti, Filippo Merloni, Michela Palleschi, Marianna Sirico, Gabriele Zoppoli, Antonino Musolino

**Affiliations:** 1Medical Oncology, Breast & GYN Unit, IRCCS Istituto Romagnolo per lo Studio dei Tumori (IRST) “Dino Amadori”, 47014 Meldola, Italy; samanta.sarti@irst.emr.it (S.S.); chiara.casadei@irst.emr.it (C.C.); lorenzo.cecconetto@irst.emr.it (L.C.); giandomenico.dimenna@irst.emr.it (G.D.M.); alberto.farolfi@irst.emr.it (A.F.); caterina.gianni@irst.emr.it (C.G.); marita.mariotti@irst.emr.it (M.M.); filippo.merloni@irst.emr.it (F.M.); michela.palleschi@irst.emr.it (M.P.); marianna.sirico@irst.emr.it (M.S.); antonino.musolino@irst.emr.it (A.M.); 2Breast Cancer Unit, Department of Cancer Medicine, Gustave Roussy, 94800 Villejuif, France; alessandro.viansone@gustaveroussy.fr; 3Gruppo Oncologico Italiano di Ricerca Clinica (GOIRC), 43100 Parma, Italy; gabriele.zoppoli@unige.it; 4Biostatistics and Clinical Trials Unit, IRCCS Istituto Romagnolo per lo Studio dei Tumori (IRST) “Dino Amadori”, 47014 Meldola, Italy; 5Department of Medicine and Surgery, University of Parma, 43121 Parma, Italy; 6Department of Internal Medicine and Medical Specialties, University of Genoa, IRCCS Ospedale Policlinico San Martino, 16132 Genova, Italy; 7Department of Medical and Surgical Sciences, University of Bologna, 40126 Bologna, Italy

**Keywords:** pharmacoprevention, breast cancer, SERMs, SERDs

## Abstract

Breast cancer remains the most common cancer in women worldwide, and new cases continue to rise in many countries. Medicines that block estrogen activity, such as tamoxifen, raloxifene, anastrozole, and exemestane, have been shown in large clinical trials to reduce the risk of developing breast cancer in women at increased risk. However, very few eligible women choose to take these preventive medicines. This review explains why prevention is important, what we currently know about the benefits and risks of available medicines, and why their use remains so limited in everyday practice. We also highlight new strategies being studied, such as using lower doses, alternative schedules, digital tools to support decision making, and the development of new medicines and vaccines. By summarizing evidence and emerging approaches, this review aims to support more informed choices and encourage wider adoption of breast cancer prevention.

## 1. Introduction: Why Prevention Matters in Modern Oncology

Over the past century, advances in cancer biology have profoundly shaped therapeutic strategies, ranging from surgery, radiotherapy, and cytotoxic chemotherapy to targeted agents, immunotherapies, and antibody-drug conjugates [1]. These milestones illustrate how advances in tumor genetics and the understanding of cancer hallmarks have progressively shaped the shift toward precision medicine [2]. Within this broader landscape, breast cancer has served as the paradigm cancer of scientific progress in oncology [3].

Breast cancer is the most frequently diagnosed malignancy and the leading cause of cancer-related death among women worldwide, accounting for over 2.3 million new cases and nearly 700,000 deaths in 2020 alone [3,4]. Over recent years, breast cancer incidence has been rising globally, with GLOBOCAN 2022 data showing increases of 1–5% annually in about half of the countries examined [5]. Despite substantial advances in early detection and treatment, primary prevention remains a critical pillar in reducing the global burden of this disease. Among the available strategies, pharmacoprevention represents a scientifically validated yet underutilized option for women at increased risk of breast cancer. Pharmacoprevention refers to the use of natural or synthetic agents to inhibit delay, or reverse carcinogenesis. In the context of breast cancer, several randomized clinical trials over the past three decades have demonstrated that selective estrogen receptor modulators (SERMs) such as tamoxifen and raloxifene, as well as aromatase inhibitors (AIs) including exemestane and anastrozole, can significantly reduce the incidence of hormone receptor-positive breast cancer in high-risk populations [6,7,8,9,10,11,12]. These findings led to the regulatory approval of tamoxifen and raloxifene by the U.S. Food and Drug Administration (FDA) for risk reduction in selected women.

### From Evidence to Practice: The Uptake Gap

Despite this robust evidence base, the clinical uptake of pharmacoprevention remains disappointingly low. Several factors contribute to this gap, including difficulties in risk stratification, limited awareness among patients and physicians, concerns over adverse effects, and a lack of systemic strategies to incorporate prevention into routine care [11,12,13]. Moreover, the identification of subgroups most likely to benefit from pharmacoprevention—and to tolerate them—remains a critical issue. This review provides a comprehensive overview of the current landscape of pharmacoprevention in breast cancer. We summarize the available agents, indications, efficacy, and safety profiles; examine factors affecting clinical implementation; and explore emerging directions, including biomarker-based strategies and novel agents under investigation. By consolidating the existing evidence and highlighting gaps in clinical translation, we aim to inform future efforts to optimize and expand the role of pharmacoprevention in breast cancer control.

## 2. Established Pharmacological Agents: The Strongest Current Evidence (SERMs and AIs)

Several pharmacological agents have proved effective in reducing the incidence of hormone receptor-positive breast cancer in women at elevated risk. The most extensively studied classes include selective estrogen receptor modulators (SERMs) and aromatase inhibitors (AIs). Their clinical development has been supported by large, randomized trials, meta-analyses, and long-term follow-up data, which have informed current clinical guidelines. Table 1 summarizes the major phase III randomized trials of breast cancer prevention, presenting absolute risk reductions together with the corresponding numbers needed to treat and to harm.

### 2.1. Selective Estrogen Receptor Modulators (SERMs)

Tamoxifen was the first pharmacologic agent to demonstrate a statistically significant preventive effect against breast cancer in women at elevated risk. Its efficacy was conclusively demonstrated in the landmark National Surgical Adjuvant Breast and Bowel Project (NSABP) P-1 trial, which enrolled over 13,000 women with a projected 5-year risk of developing breast cancer of at least 1.66%. Participants were randomized to receive either tamoxifen 20 mg/day or placebo for five years. The trial demonstrated a 49% relative reduction in the incidence of invasive breast cancer among women assigned to tamoxifen, with the greatest protective effect observed in estrogen receptor-positive (ER+) tumors, consistent with its anti-estrogenic mechanism of action [6]. However, despite its proven efficacy, tamoxifen use was associated with significant adverse effects, most notably an increased incidence of endometrial cancer and thromboembolic complications such as deep vein thrombosis and pulmonary embolism [13,19,21]. These risks were particularly pronounced in women over 50 years of age, necessitating careful selection of candidates for tamoxifen-based pharmacoprevention and close monitoring during treatment. The efficacy and long-term benefits of tamoxifen were further substantiated by subsequent large-scale randomized trials, including the International Breast Cancer Intervention Study I (IBIS-I) and the Royal Marsden Hospital tamoxifen pharmacoprevention trial [14,22]. These studies confirmed tamoxifen’s role in significantly reducing the incidence of ER+ breast cancers, thus solidifying its place as the first-line agent in pharmacoprevention for premenopausal and select postmenopausal women at high risk.

Raloxifene, a second-generation selective estrogen receptor modulator (SERM), was originally developed and approved for the prevention and treatment of osteoporosis in postmenopausal women due to its estrogen agonist effects on bone and lipid metabolism. Its potential role in breast cancer prevention emerged from its anti-estrogenic activity in breast tissue, analogous to tamoxifen but with a more favorable toxicity profile. The Study of Tamoxifen and Raloxifene (STAR) trial, a large randomized, double-blind, phase III trial conducted by the National Surgical Adjuvant Breast and Bowel Project (NSABP), directly compared raloxifene (60 mg/day) and tamoxifen (20 mg/day) over five years in more than 19,000 postmenopausal women at increased risk for breast cancer. The trial demonstrated that raloxifene was nearly as effective as tamoxifen in reducing the incidence of invasive breast cancer, particularly ER-positive tumors, which account for the majority of hormone-driven breast cancers in this population [7,8]. Importantly, raloxifene exhibited a significantly more favorable safety profile compared to tamoxifen. Women treated with raloxifene had substantially lower rates of endometrial cancer, thromboembolic events (including pulmonary embolism and deep vein thrombosis), and cataract formation [7,19]. Although tamoxifen demonstrated slightly greater efficacy in reducing noninvasive breast cancers such as ductal carcinoma in situ (DCIS), the overall benefit-risk profile of raloxifene was considered superior for many postmenopausal candidates. As a result, raloxifene emerged as the preferred preventive agent in postmenopausal women who are concerned about the adverse effects of tamoxifen or have contraindications to its use. Its dual utility in maintaining bone density while reducing breast cancer risk offers an additional advantage in this population, many of whom are also at risk for osteoporosis [8,13,22].

### 2.2. Aromatase Inhibitors (AIs)

Aromatase inhibitors (AIs) represent a distinct class of endocrine agents that function by irreversibly or competitively inhibiting the aromatase enzyme, which catalyzes the conversion of androgens into estrogens in peripheral tissues. This mechanism is particularly relevant in postmenopausal women, in whom ovarian estrogen production ceases and peripheral aromatization becomes the primary source of circulating estrogens. By significantly reducing systemic estrogen levels, AIs exert a potent anti-estrogenic effect, thereby lowering the risk of estrogen receptor-positive (ER+) breast cancer. The preventive potential of AIs has been robustly evaluated in two pivotal randomized controlled trials. The IBIS-II trial (International Breast Cancer Intervention Study II) enrolled more than 3800 postmenopausal women at elevated risk of breast cancer based on family history or prior benign proliferative breast disease. Participants were assigned to receive anastrozole 1 mg/day or placebo for five years. Results demonstrated a 53% reduction in the incidence of breast cancer, with the preventive effect almost exclusively confined to ER+ tumors [10,11]. The protective benefit of anastrozole was consistent across subgroups and was not associated with an increased risk of cardiovascular events or thromboembolism, distinguishing it favorably from SERMs. The MAP.3 trial (Mammary Prevention 3), conducted by the National Cancer Institute of Canada Clinical Trials Group, similarly evaluated the steroidal AI exemestane at 25 mg/day in over 4500 healthy postmenopausal women with at least one major risk factor for breast cancer. The study reported a 65% relative risk reduction in invasive breast cancer compared to placebo [11,12]. Notably, exemestane’s preventive efficacy was evident early in the treatment course and sustained throughout the intervention period. Like anastrozole, exemestane exhibited no significant increase in thromboembolic or endometrial events, reinforcing the favorable safety profile of AIs in the prevention setting. However, AIs are not devoid of adverse effects. Their profound suppression of estrogen can lead to musculoskeletal symptoms (arthralgia, myalgia), accelerated bone mineral density loss, and vasomotor disturbances (e.g., hot flashes), which may negatively impact quality of life and adherence to therapy. These side effects are particularly relevant in older women who may already be at risk for osteoporosis or joint disorders. Consequently, baseline and periodic bone density assessments, calcium and vitamin D supplementation, and the potential use of bone-protective agents such as bisphosphonates or denosumab are essential components of comprehensive care in patients undergoing AI-based prevention [13,21].

### 2.3. Long-Term Efficacy and Safety Considerations of SERMS and AIs

The long-term effectiveness of SERMs and AIs in preventing ER+ breast cancer is well substantiated by extended follow-up data from multiple trials. For instance, in the IBIS-I trial, the protective benefit of tamoxifen persisted for up to two decades following the initial five-year treatment period, underscoring the capacity of endocrine prevention to achieve durable reductions in breast cancer risk [14,22]. Similarly, follow-up analyses from IBIS-II and MAP.3 are beginning to provide insights into the sustained efficacy of AIs beyond the treatment window [10,12]. Despite the compelling efficacy data, real-world implementation of pharmacologic prevention remains suboptimal, largely due to issues with long-term adherence. Adverse effects, especially menopausal symptoms and fears about long-term toxicity, often lead to early discontinuation of therapy [13,19,21]. This phenomenon underscores the importance of shared decision-making, risk-benefit counseling, and patient education, which are critical in setting expectations and improving adherence. Another emerging consideration is the potential role of intermittent or tailored dosing strategies to balance efficacy with tolerability, although such approaches remain investigational [19,23].

Both SERMs and AIs offer substantial, evidence-based reductions in breast cancer incidence among women with elevated risk, particularly those with ER+ disease. Their incorporation into national and international guidelines reflects decades of rigorous clinical research. However, translating this efficacy into real-world effectiveness requires individualized clinical decision-making, proactive management of side effects, and ongoing support to sustain adherence. Expanding the uptake of pharmacologic prevention will depend not only on refining agents and indications, but also on optimizing communication between clinicians and patients regarding the long-term benefits and manageable risks of these interventions [6,7,12].

## 3. Risk Stratification and Identification of Eligible Populations

The successful implementation of pharmacological breast cancer prevention depends critically on the accurate identification and stratification of women at increased risk. Effective risk stratification not only determines eligibility for pharmacoprevention but also supports shared decision-making, enabling clinicians to tailor preventive strategies based on individual benefit-risk profiles. This is especially important given the long-term nature of preventive therapies and their potential adverse effects.

### 3.1. Quantitative Risk Assessment Models

Several validated models are available to estimate an individual woman’s risk of developing breast cancer (Table 2). The most widely used include:

Gail model (Breast Cancer Risk Assessment Tool), which estimates 5-year and lifetime risk based on age, reproductive history, family history of breast cancer, history of breast biopsies, and race/ethnicity [24,25]. A 5-year risk threshold of ≥1.66% has traditionally been used to define eligibility for tamoxifen in U.S. guidelines [26]. Although simple and easy to use, the Gail model tends to underestimate risk in women with a strong family history or hereditary syndromes and does not incorporate genetic mutations.Tyrer-Cuzick (IBIS) model, which incorporates additional risk factors such as body mass index, hormone replacement therapy, age at menarche and menopause, and more detailed family history including BRCA1/2 status [27,28]. This model provides both 10-year and lifetime risk estimates and is considered more comprehensive, especially for identifying women with hereditary breast cancer risk. Its integration with mammographic density data further enhances its discriminatory accuracy [29].BOADICEA (Breast and Ovarian Analysis of Disease Incidence and Carrier Estimation Algorithm) model, which is developed to estimate the probability of being a carrier of pathogenic variants in genes such as BRCA1, BRCA2, PALB2, CHEK2, and ATM, in addition to calculating breast and ovarian cancer risk [30,31]. With its increasing availability and integration in genetic counseling tools, BOADICEA is gaining traction as a precision risk stratification model.

### 3.2. Clinical Criteria for High-Risk Populations

Women considered at elevated risk for breast cancer may be eligible for preventive pharmacological interventions based on several well-established criteria. In the United States, a 5-year risk of invasive breast cancer ≥ 1.66%, as calculated by validated models such as the Gail model, is commonly used as a threshold for pharmacoprevention eligibility [24,26]. In contrast, European guidelines—such as those from NICE (UK) and ESMO—advocate for a more conservative threshold of ≥3% over 5 years, aiming to more precisely identify women at very high risk who may derive greater benefit from agents like selective estrogen receptor modulators (SERMs) or aromatase inhibitors (AIs) [32,33,34].

In addition to model-based risk estimates, clinical guidelines also define high-risk status through specific clinical and pathological factors. Histologically confirmed high-risk lesions, such as atypical ductal hyperplasia (ADH), atypical lobular hyperplasia (ALH), or lobular carcinoma in situ (LCIS), are associated with a 4 to 10-fold increased risk of subsequent invasive cancer [35]. A strong family history of breast cancer, particularly when involving first-degree relatives with early-onset, bilateral, or male breast cancer, further contributes to elevated risk. Genetic predisposition is another major determinant, with pathogenic variants in BRCA1, BRCA2, and other moderate to high-penetrance genes such as PALB2 and CHEK2 conferring substantially increased susceptibility [30,31]; however, in BRCA1 mutation carriers, the utility of SERMs and aromatase inhibitors remains limited given the predominantly ER-negative phenotype of associated tumors [31]. Finally, a history of chest radiation before the age of 30—such as treatment administered for Hodgkin lymphoma—significantly raises the risk of breast cancer, typically manifesting after a latency of 10–15 years [36].

It is critical to differentiate which subtypes of breast cancer are impacted by pharmacologic prevention. Agents such as tamoxifen and aromatase inhibitors predominantly reduce the incidence of estrogen receptor-positive (ER+) tumors, with limited or no benefit for ER-negative cancers [22,24,25]. This reinforces the need to tailor pharmacoprevention strategies according to both tumor biology and the patient’s individual risk profile.

### 3.3. Emerging Tools for Personalized Stratification

Ongoing research is refining risk prediction through the integration of novel biomarkers and technological tools. Polygenic Risk Scores (PRS), which aggregate the small effects of hundreds to thousands of common single-nucleotide polymorphisms (SNPs), provide a scalable method to stratify risk across populations and have shown added predictive value when combined with classical risk models, particularly in women without strong family histories [37,38]. Mammographic breast density, a well-established independent risk factor, not only influences cancer risk but also affects mammographic sensitivity, and its presence may prompt consideration of preventive interventions and adjunct screening modalities [26,29]. In parallel, artificial intelligence and machine learning algorithms—trained on imaging, demographic, and clinical data—are being validated for their capacity to predict both short- and long-term breast cancer risk with greater precision, offering the potential for real-time individualized prediction and seamless integration into clinical decision support systems [39,40,41].

## 4. Guidelines Recommendations

Substantial evidence supports pharmacoprevention of hormone receptor-positive breast cancer, and several international organizations have issued clinical guidelines to help clinicians identify eligible women and select appropriate agents [9,10,11,12,32,33,34,42,43,44]. Table 3 summarizes key international recommendations for risk-reducing pharmacologic interventions.

The United States Preventive Services Task Force (USPSTF) gives a Grade B recommendation for offering tamoxifen, raloxifene, or aromatase inhibitors (AIs) to women at increased risk with a low likelihood of serious adverse effects, reflecting moderate certainty of a moderate net benefit; conversely, a Grade D recommendation advises against use in women at average or low risk. USPSTF emphasizes individualized risk–benefit assessment and shared decision-making [42].

The American Society of Clinical Oncology (ASCO, 2019) reaffirms pharmacologic prevention for appropriately selected high-risk women. Tamoxifen is recommended for both pre- and postmenopausal women, whereas raloxifene and AIs (exemestane, anastrozole) apply only to postmenopausal women. Choice of agent should incorporate patient preferences, menopausal status, comorbidities (e.g., thromboembolic risk, osteoporosis), and absolute risk, with counseling ideally delivered in specialized high-risk settings and supported by decision aids [43].

The National Comprehensive Cancer Network (NCCN) identifies candidates for pharmacoprevention including women with a 5-year invasive breast cancer risk ≥ 1.7% by the Gail model (with other validated models such as Tyrer-Cuzick informing clinical judgment), those with high-risk histology (ADH, ALH, LCIS), strong family history, or pathogenic variants in susceptibility genes. Tamoxifen is preferred for premenopausal women; in postmenopause, tamoxifen, raloxifene, or AIs are options. Raloxifene may be preferred over tamoxifen when endometrial risk is a concern, but both SERMs increase VTE risk and should be avoided in women at high thromboembolic risk; AIs require bone-health monitoring. NCCN encourages integrating pharmacologic prevention with surveillance, lifestyle interventions, and genetic counseling where appropriate [43].

The European Society for Medical Oncology (ESMO) supports tamoxifen for high-risk pre- and postmenopausal women, with raloxifene or AIs as alternatives for postmenopausal women unsuitable for tamoxifen. ESMO also highlights European-specific barriers (e.g., heterogeneous policies, limited awareness, medico-legal concerns) and advocates clinician education, high-risk multidisciplinary clinics, and improved patient communication to increase uptake [33].

In the United Kingdom, the National Institute for Health and Care Excellence (NICE) recommends tamoxifen or raloxifene for five years in women at moderate (17–29% lifetime) or high (≥30%) risk, with aromatase inhibitors (e.g., anastrozole or exemestane) as alternatives for postmenopausal women intolerant to SERMs. Risk is typically assessed with Tyrer-Cuzick and may be supplemented by genetic testing or specialist evaluation; bone health monitoring is advised during AI therapy, alongside follow-up to support adherence, side-effect management, lifestyle change, screening, and genetic counseling [32].

## 5. Why Uptake Remains Low: Benefit-Risk Balance and Cost-Effectiveness

Any pharmacological intervention for primary prevention must demonstrate not only clinical efficacy, but also a favorable balance of benefits versus risks, and ideally, cost-effectiveness in the broader healthcare context. For breast cancer prevention, this evaluation is particularly nuanced, given that interventions are offered to asymptomatic women and the absolute benefit varies according to individual risk levels.

### 5.1. Efficacy vs. Adverse Effects

Both SERMs and AIs significantly reduce the incidence of estrogen receptor-positive breast cancer in high-risk populations, but their use is associated with non-negligible side effects. Tamoxifen reduces the incidence of estrogen receptor-positive (ER+) breast cancer by approximately 49–50%, with consistent preventive efficacy demonstrated across different risk groups. In the NSABP P-1 trial, tamoxifen reduced the incidence of invasive breast cancer from 43 to 22 cases per 1000 women over five years (risk ratio [RR] 0.51; *p* < 0.00001) [5]. Women with atypical hyperplasia derive the greatest absolute benefit, with risk reductions approaching 86% [19]. The number needed to treat (NNT) varies significantly by baseline risk: in women with atypical hyperplasia, the NNT can be as low as 28, whereas in women at average risk, the NNT often exceeds 100–200, reflecting the limited absolute benefit in this population [5,19]. Despite its efficacy, tamoxifen use is associated with well-characterized adverse events, including an increased risk of endometrial cancer (particularly in postmenopausal women), venous thromboembolism (VTE), and ocular toxicities such as cataracts [5,12,18,19,45,46,47,48]. These risks, which are dose-dependent, represent significant limitations to widespread implementation, especially in asymptomatic individuals undergoing preventive therapy.

In postmenopausal women, tamoxifen is associated with up to a 2–4 fold increased risk of endometrial cancer. By contrast, in premenopausal women (<50 years), this increase is not statistically significant (RR ~1.19; 95% CI 0.53–2.65; *p* = 0.6), based on trials involving 20 mg/day for five years [18,21]. Tamoxifen doubles the risk of deep vein thrombosis during active treatment (RR 2.3, 95% CI 1.23–4.31; *p* = 0.009). Pulmonary embolism risk is increased but not significantly so, and risks notably decrease post treatment [5,18,21,48].

Raloxifene, which is approved for breast cancer prevention in postmenopausal women, demonstrates slightly reduced efficacy compared to tamoxifen. In the STAR trial (Study of Tamoxifen and Raloxifene), raloxifene achieved approximately a 38% reduction in invasive ER-positive breast cancer incidence, compared to 49% for tamoxifen [6,7]. The number needed to treat (NNT) with raloxifene is approximately 29–30 in high-risk populations, closely comparable to tamoxifen in similar cohorts [6,8]. However, raloxifene offers a superior safety profile. Unlike tamoxifen, raloxifene lacks estrogen agonist activity on the endometrium, virtually eliminating the risk of endometrial cancer [7,8,48]. Moreover, raloxifene is associated with a lower incidence of venous thromboembolic events compared to tamoxifen, although the risk remains elevated relative to untreated populations [6,8]. These safety advantages make raloxifene an attractive option for postmenopausal women, particularly those at heightened risk for endometrial pathology or with preexisting thrombotic risk factors. Cost-effectiveness analyses support raloxifene as the preferred preventive agent for older postmenopausal women or those contraindicated for tamoxifen, due to its favorable risk-benefit profile [6,8,49]. These conclusions are primarily supported by data from the STAR trial, including its long-term follow-up analyses [6,8].

Aromatase inhibitors such as anastrozole and exemestane show risk reductions of up to 60–70%, the highest among pharmacologic agents for breast cancer prevention. However, they are associated with musculoskeletal pain, hot flashes, arthralgia, and a significant decline in bone mineral density, which can lead to increased fracture risk [9,11,48]. In the IBIS-II trial, anastrozole showed a favorable NNT of 36 over five years in high-risk postmenopausal women, although tolerability concerns remain a limiting factor in widespread adoption [9,10].

In comparative modeling analyses, the NNT to prevent one case of breast cancer varies substantially with risk level and type of lesion. For example, women with lobular carcinoma in situ or atypical hyperplasia derive the greatest absolute benefit. However, this must be balanced with the number needed to harm (NNH) for serious side effects such as thromboembolism or osteoporosis-related fractures [9,10,25]. Tamoxifen, for instance, has an NNH for pulmonary embolism ranging from 100–200 depending on age and menopausal status [5,24,31,48]. Personalized assessment of absolute benefit versus potential harm is essential to optimize patient outcomes and ensure appropriate selection of preventive agents.

### 5.2. Ethical and Policy Implications

Because preventive agents are offered to healthy individuals, the ethical threshold for harm is lower than in therapeutic settings. Informed consent, transparent communication of risks, and continuous monitoring for adverse events are critical. Policymakers must also weigh the societal value of reducing breast cancer incidence, downstream treatment costs, and the emotional and economic burden on patients and families [18,48,49,50].

### 5.3. Quality of Life Considerations

The impact of side effects on health-related quality of life (HRQoL) plays a central role in determining both the acceptability and long-term adherence to breast cancer pharmacoprevention. Although randomized controlled trials have consistently shown that selective estrogen receptor modulators (SERMs) and aromatase inhibitors (AIs) significantly reduce the incidence of estrogen receptor–positive (ER+) breast cancer, real-world data reveal that adverse effects often undermine their sustained use [9,10,50].

Common symptoms—such as hot flashes, arthralgia, myalgia, fatigue, and mood disturbances—are particularly frequent with AIs and contribute to therapy discontinuation [9,10,25]. For instance, in a longitudinal study of women taking tamoxifen, nearly 30% discontinued treatment prematurely due to side effects, despite being at elevated risk [50,51]. Similarly, in the IBIS-II trial, patients on anastrozole reported higher rates of musculoskeletal pain and bone loss, which negatively impacted adherence [9,10].

Evidence from observational studies and patient-reported outcome measures further highlights that even mild-to-moderate symptoms can lead to non-adherence, especially when the perceived benefit is unclear or poorly communicated [50,51,52]. The psychological burden of taking a medication to prevent a disease that may never occur—combined with immediate and tangible side effects—represents a major challenge in clinical practice [50,51,52]. To support adherence and minimize HRQoL impact, effective risk communication, tailored counseling, and ongoing follow-up are essential. Tools such as decision-analytic models incorporating quality-adjusted life years (QALYs) help quantify the trade-off between benefit and harm [49,53]. These models frequently demonstrate that pharmacoprevention is cost-effective, or even cost-saving, in women at high absolute risk. Conversely, in women with lower risk, the cost-effectiveness decreases, due to a smaller absolute benefit and proportionally greater burden of side effects [49,53].

These findings underscore the importance of accurate risk stratification, shared decision-making, and personalized communication to optimize both clinical outcomes and patient experience in pharmacoprevention [49,50,51,53].

## 6. Why Uptake Remains Low: Barriers to Clinical Implementation

Despite compelling evidence supporting the efficacy of SERMs and AIs in preventing hormone receptor-positive breast cancer, the clinical uptake of pharmacoprevention remains disappointingly low. Estimates from population-based studies suggest that fewer than 5% of eligible women are offered or initiate preventive therapy [51,52,53,54]. Multiple and interrelated factors contribute to this underutilization, including challenges related to provider behavior, patient perceptions, systemic healthcare limitations, and regulatory or cultural contexts [51,52,53,54,55].

### 6.1. Physician-Related Barriers

A deeper examination reveals that a significant proportion of primary care providers lack formal training in oncologic risk communication. Continuing medical education (CME) programs rarely emphasize preventive oncology, leading to gaps in awareness and clinical confidence. Additionally, providers may overestimate the side effects or underestimate the benefits of pharmacoprevention, perpetuating clinical inertia [13,18,21,54]. The fragmentation between primary care and oncology services often results in missed opportunities for early intervention, especially in the absence of robust referral protocols [18,54].

Numerous studies have shown that healthcare providers are often reluctant to initiate discussions about pharmacoprevention. Primary care physicians may feel uncomfortable interpreting breast cancer risk models, lack familiarity with preventive agents, or perceive pharmacoprevention as the domain of specialists [13,18,21,55]. Oncologists, although knowledgeable, are more likely to focus on treatment rather than prevention, especially in the absence of institutional pathways that support proactive risk identification [18,56]. Time constraints, competing clinical priorities, and a lack of reimbursement for preventive counseling further reduce the likelihood that providers will engage in risk assessment or prescribe preventive agents [13,18,21]. Expanded research shows that many patients interpret the term ‘pharmacoprevention’ as suggestive of disease presence, creating psychological resistance. Fear of stigmatization or being perceived as ‘ill’ may deter otherwise eligible candidates. Moreover, studies show that socioeconomic status, race, and educational level correlate with uptake, suggesting that underserved populations may face compounded barriers due to access, health literacy, and provider bias [13,18,21,51]. Behavioral economics models indicate that immediate costs or discomforts often outweigh future benefits in patient decision-making [13,21,52].

### 6.2. Patient-Related Barriers

From the patient’s perspective, concerns about side effects are a major deterrent. The risk of venous thromboembolism, endometrial cancer (for SERMs), or bone loss and arthralgia (for AIs) often outweighs perceived benefits, especially in asymptomatic individuals. Furthermore, many women are unaware of their personal breast cancer risk or the existence of preventive options [13,18,21]. Misperceptions about pharmacoprevention, including the belief that it equates to “taking chemotherapy without having cancer”, may fuel resistance. Cultural beliefs, mistrust of medications, or prior negative experiences with hormonal treatments (e.g., HRT) can further erode acceptance [52,53,54]. Beyond structural inertia, the lack of institutional investment in risk stratification tools—such as integrated risk calculators within electronic health systems—hampers proactive prevention strategies. Reimbursement models often prioritize treatment over prevention, disincentivizing healthcare providers from investing time and resources in identifying and counseling at-risk individuals [54,56]. Moreover, in resource-limited settings, competing healthcare priorities and budgetary constraints frequently result in the underfunding of preventive care infrastructure [56,57].

Adherence is another key challenge. Even among women who initiate pharmacoprevention, discontinuation rates approach 30–40% within the first year, driven by intolerability or doubts about efficacy [18,51,58].

### 6.3. System-Level and Structural Challenges

Systemic issues also play a critical role in limiting access to pharmacoprevention, including the lack of routine risk stratification in primary care, the absence of standardized protocols to guide referral and management of high-risk women, the insufficient integration of breast cancer prevention into electronic health records and decision-support systems, and the limited coverage or reimbursement for preventive agents in certain healthcare systems [18,55,56]. In many countries, pharmacoprevention is still not considered a public health priority, and preventive interventions are rarely incentivized at the financial or institutional level [18,56]. Promising strategies include the use of automated EHR prompts to identify high-risk women during routine visits, which can substantially increase preventive counseling rates. The involvement of multidisciplinary teams—such as primary care physicians, genetic counselors, and nurse navigators—may further streamline care pathways [13,18,56,57,59]. Moreover, public health campaigns that frame breast cancer prevention similarly to vaccination or smoking cessation initiatives could help normalize pharmacoprevention [13,52,53,60]. Finally, strategic partnerships with advocacy groups, professional societies, and policymakers will be critical to generate momentum for widespread implementation [18,56,61].

### 6.4. Legal, Ethical, and Cultural Considerations

Offering pharmacoprevention to healthy individuals raises specific ethical considerations. The balance of risks and benefits must be individualized, and informed consent should be emphasized [39,54]. In populations with limited health literacy or language barriers, effective communication becomes a crucial determinant of uptake [50,52,53]. As previously discussed, regulatory labeling and national guidelines vary widely across countries. For example, tamoxifen and raloxifene are FDA-approved for risk reduction in the U.S. but may lack formal indications or reimbursement in European countries, thereby limiting access [5,6,32,38].

### 6.5. Strategies to Improve Uptake

To overcome these barriers, multifaceted strategies are needed, including education and training for primary care providers and gynecologists on breast cancer risk and preventive options, the development of patient-centered decision aids and communication tools to facilitate shared decision-making, and the incorporation of risk prediction algorithms into electronic health records with automated alerts for high-risk women. Policy initiatives aimed at expanding insurance coverage and incentivizing preventive care are also crucial. Notably, pilot programs that embed prevention into routine care—such as high-risk breast clinics or genetic counseling services—have shown promise in increasing both the prescription and acceptance of preventive agents [11,12,59].

Emerging Strategies: Digital Health Tools and Personalized Decision Aids. In the context of limited real-world uptake and adherence to pharmacological prevention with SERMs and AIs, digital health interventions (including web-based risk calculators, mobile applications, and tailored decision aids) are emerging as complementary solutions to support women in understanding their personal risk and making informed choices about preventive therapy. These tools enable individualized risk visualization using models such as Gail, BCSC, or Tyrer-Cuzick, and provide comparative data on the benefits and harms of preventive agents. Several decision aids, including RealRisks, BreastHealthDecisions.org, and the WISDOM e-tool, have been evaluated in clinical studies and have been consistently shown to improve knowledge, reduce decisional conflict, and increase willingness to consider preventive therapy [26,53,62,63,64]. By integrating clinical and demographic data, these platforms generate personalized risk scores that enhance women’s perception of their risk and promote value-congruent decisions. The experience with RealRisks in the Columbia University primary care network, for example, demonstrated increased provider-patient discussions about pharmacoprevention, particularly among Hispanic women, a group often underrepresented in preventive interventions [53,62,63].

Beyond risk communication, digital decision aids have also been shown to influence patient knowledge and attitudes. Tools such as the WISDOM e-tool, especially when implemented in primary care or high-risk screening settings, not only improve risk perception but also foster meaningful discussions about prevention. Importantly, interventions co-designed with women through participatory approaches have demonstrated greater usability, clinical relevance, and engagement [62,64]. Embedding these digital instruments into electronic medical records (EMRs) can further facilitate systematic identification of high-risk women. The PROMPT trial, for instance, showed that EMR-based alerts and provider-focused digital interventions significantly improved the offer rates of pharmacoprevention, while initiatives such as the BC-Predict pilot highlight the feasibility of incorporating risk tools into national screening programs [56,57,59]. Ensuring interoperability with existing clinical workflows remains essential to guarantee sustainability.

Digital health also holds promise for adherence support. Mobile health (mHealth) solutions, including smartphone applications and SMS-based platforms, can provide medication reminders, track side effects and patient-reported outcomes (PROMs), and enable remote communication with healthcare teams. PROMs platforms, already piloted in oncology, may be adapted to monitor adherence and persistence in preventive contexts, thus supporting both short- and long-term outcomes [56,57,58,64]. Yet, challenges persist, such as digital literacy gaps among older or underserved populations, the limited integration of tools into EMRs, and regulatory or reimbursement barriers. Addressing equity is critical: interventions should incorporate multilingual support, culturally sensitive content, and adaptable formats to ensure accessibility for all risk-eligible women. Looking ahead, next-generation tools are rapidly evolving toward adaptive, intelligent systems. Artificial intelligence and machine learning approaches can enhance risk stratification by incorporating genomic, lifestyle, and behavioral data, and dynamically updating risk scores as patient characteristics change (e.g., menopausal status, BMI, family history). Early-phase projects are also exploring AI-assisted decision coaching capable of adapting to patients’ learning preferences and emotional states [34,35].

Finally, policy frameworks may accelerate the adoption of digital health tools for breast cancer pharmacoprevention. Germany’s Digital Health Act (DVG) already reimburses selected evidence-based digital health applications (DiGAs), and similar approaches are being considered in France and the Netherlands [60,61,65]. These developments suggest that digital decision aids and mobile health solutions could soon become integral components of formal preventive care pathways, complementing pharmacological strategies and addressing some of the barriers that have historically limited their implementation.

## 7. Emerging Strategies: Low-Dose Tamoxifen

One of the most promising strategies to improve the tolerability, acceptability, and adherence to endocrine pharmacoprevention is the use of low-dose tamoxifen (LDT). While the standard 20 mg/day dose has demonstrated robust efficacy in reducing the incidence of hormone receptor–positive breast cancer, its broader adoption in the primary prevention setting has been hampered by concerns regarding safety, particularly the risks of thromboembolic events, endometrial cancer, vasomotor symptoms, and ocular toxicities in postmenopausal women [24,30,39]. These adverse effects have limited its uptake to a minority of eligible candidates, underscoring the need for safer and more acceptable approaches. The landmark TAM-01 trial, a randomized, double-blind, placebo-controlled phase III study, addressed this limitation by testing a 75% dose reduction (5 mg/day) in women with a history of ductal carcinoma in situ (DCIS), atypical ductal or lobular hyperplasia, or lobular carcinoma in situ (LCIS). In over 500 women, TAM-01 demonstrated a 52% reduction in recurrent breast events, including both invasive and non-invasive lesions, at a median 5-year follow-up, without a significant increase in serious adverse events such as endometrial carcinoma or thromboembolism [20,65]. Subsequent 10-year follow-up confirmed that the protective effect of LDT persisted well beyond treatment discontinuation, with recurrence rates remaining approximately 50% lower than placebo [20]. These results, which are also summarized in Table 1 alongside absolute risk reductions and numbers needed to treat, reinforce the favorable benefit–risk profile of low-dose tamoxifen as compared to standard-dose therapy.

Complementary approaches are also under investigation, including a phase II trial (NCI 2022 02973) comparing oral tamoxifen (5–10 mg/day) with topical afimoxifene (4-OHT) gel in women with atypical ductal hyperplasia or LCIS. The topical formulation aims to achieve high local breast tissue concentrations with minimal systemic exposure, thereby preserving efficacy while reducing vasomotor and systemic effects. Early pharmacokinetic data support this rationale, showing significant modulation of estrogen receptor signaling and proliferation markers (e.g., Ki-67) in breast tissue, with fewer systemic toxicities [66]. Results of the ongoing trial will further clarify the feasibility, tolerability, and adherence of this novel administration route [67].

## 8. Emerging Strategies: Ongoing Clinical Trials

Building on the promising results of the TAM-01 trial and subsequent investigations with low-dose tamoxifen, multiple ongoing clinical studies are now testing whether dose de-escalation and alternative delivery strategies can further improve tolerability and adherence in preventive settings. Catalogued in the National Cancer Institute (NCI) Clinical Trials Search platform, these trials collectively aim to reduce toxicity, enhance quality of life, and tailor interventions to individual risk and tolerance profiles. Among them, the BABY-TEARS trial (NCT06364267) is evaluating ultra-low-dose regimens by comparing tamoxifen 10 mg every other day with exemestane 25 mg every other day in postmenopausal women at high risk for breast cancer. In addition to assessing menopause-specific quality of life (MENQOL), the trial incorporates biomarker endpoints such as Ki-67 and mammographic density, with enrollment scheduled to begin in September 2025 [68]. The TOLERANT trial (NCT06033092), co-funded by the European Commission through Italy’s National Recovery and Resilience Plan (PNRR), explores a multimodal approach combining low-dose tamoxifen with lifestyle interventions such as intermittent caloric restriction and structured physical activity. This four-arm phase II study evaluates sex hormone-binding globulin (SHBG) modulation as the primary endpoint, alongside secondary markers including inflammation, metabolism, microbiome, and mammographic density. Importantly, it integrates digital adherence tools and patient-reported outcomes (PROs), exemplifying a next-generation, patient-centered model of pharmacoprevention [69].

At the phase III level, the LoTam trial (NCI 2024 06672) is testing tamoxifen 5 mg/day against standard endocrine therapies (tamoxifen 20 mg or aromatase inhibitors) in postmenopausal women with early-stage, molecularly low-risk breast cancer or a history of intraepithelial lesions. By assessing non-inferiority for recurrence prevention, as well as differences in thromboembolic and endometrial events, quality of life, adherence, and biomarker outcomes, this trial is positioned to deliver high-level evidence on the viability of dose de-escalation strategies [70]. Finally, the ELDER study investigates intermittent low-dose exemestane schedules (25 mg every other day or three times per week) in postmenopausal women with stage 0–II ER-positive breast cancer. Early results suggest that intermittent dosing can maintain estrogen suppression while improving tolerability and adherence compared with daily regimens [71].

Unlike cytotoxic chemotherapy, where suboptimal dosing may promote resistance [58], all these trials are designed to identify the minimum effective dose and duration, balancing efficacy with tolerability. They illustrate a paradigm shift toward precision prevention, emphasizing patient-centered, low-toxicity, and biomarker-guided strategies. By addressing the key limitations of current preventive agents, they may broaden the therapeutic window, increase uptake, and help integrate pharmacoprevention as a viable component of comprehensive breast cancer risk reduction [59]. Table 4 summarizes these ongoing clinical efforts, including additional early-phase trials not detailed in the text.

## 9. Emerging Strategies: Future Directions in Pharmacoprevention

Beyond trial outcomes, mechanistic and translational insights are increasingly shaping prevention strategies. As already mentioned, genomic risk tools, particularly PRS, can refine absolute risk estimates when added to clinical models and mammographic density, potentially improving identification of women most likely to benefit from endocrine prevention. Such refinement, by clarifying expected absolute benefit, may also enhance uptake and adherence to pharmacoprevention in practice [43]. In parallel, the immune microenvironment of early breast lesions, including DCIS, shows prognostic variation in tumor infiltrating lymphocytes (TILs) and checkpoint ligand expression (e.g., PD-L1), supporting the concept that immune contexture and stromal–epithelial interactions influence progression risk and may become biomarkers to tailor preventive interventions [15,72]. Lessons from other cancers underscore both promise and caveats of pharmacoprevention: aspirin reduces colorectal cancer risk in Lynch syndrome, whereas β-carotene supplementation was associated with increased lung cancer incidence in smokers. These findings emphasize the need for biologically grounded targets and careful population selection [73,74].

Recent advances in molecular oncology, pharmacogenomics, immunoprevention, and drug delivery systems are fostering the development of next-generation pharmacopreventive strategies. These approaches aim to not only expand the therapeutic armamentarium for HR+ breast cancer but also address the current unmet need for preventive interventions in hormone receptor-negative, particularly triple-negative breast cancer (TNBC) [75,76]. Table 5 summarizes emerging strategies and investigational agents representing future directions in breast cancer pharmacoprevention.

### 9.1. Selective Estrogen Receptor Degraders (SERDs)

Next-generation oral SERDs have emerged as promising candidates to overcome limitations of traditional anti-estrogen therapies. These agents induce complete degradation of the estrogen receptor (ER), including mutant forms such as ESR1, and may be effective in circumventing endocrine resistance observed in both preventive and therapeutic settings. Agents under development include the following:Elacestrant, an oral selective estrogen receptor degrader (SERD), has demonstrated significant clinical activity in ER-positive, HER2-negative, ESR1-mutant metastatic breast cancer, as shown in the EMERALD trial (NCT03778931). In this phase III study, elacestrant significantly improved progression-free survival compared to standard endocrine therapy, particularly in patients harboring ESR1 mutations, establishing it as the first oral SERD approved for use in this setting [77]. Building on these results, elacestrant is now being investigated in earlier-stage disease and prevention-oriented settings, including the ELEVATE trial (NCT05563220), a phase Ib/II umbrella study assessing elacestrant in combination with various targeted agents (e.g., CDK4/6, PI3K, and AKT inhibitors) in patients previously treated with endocrine therapy and CDK4/6 inhibitors [78]. These efforts aim to expand its potential role from metastatic disease to adjuvant and preventive applications, particularly in endocrine-sensitive tumors.Camizestrant, imlunestrant, and amcenestrant have demonstrated favorable pharmacokinetic properties and manageable safety profiles across Phase I–III trials, and ongoing studies are evaluating their role in both adjuvant and preventive settings. In the phase III EMBER-3 trial, the investigators enrolled patients with ER-positive, HER2-negative advanced breast cancer that recurred or progressed during or after aromatase inhibitor therapy, administered alone or with a CDK4/6 inhibitor. treatment with imlunestrant led to significantly longer progression-free survival than standard therapy among those with ESR1 mutations but not in the overall population. Imlunestrant-abemaciclib significantly improved progression-free survival as compared with imlunestrant, regardless of ESR1-mutation status [79,80].

Their oral administration, high bioavailability, and reduced risk of thromboembolic events compared to SERMs render them particularly attractive for long-term use in asymptomatic, high-risk individuals. Additionally, their activity in ESR1-mutant clones makes them especially relevant for women with atypical hyperplasia or lobular carcinoma in situ (LCIS), who may harbor occult genomic alterations [81].

### 9.2. Estrogen-Only Therapy

Contrary to earlier assumptions, estrogen-only therapy (ET) in postmenopausal women with prior hysterectomy has shown a paradoxical protective effect against breast cancer. A comprehensive meta-analysis of 21 randomized trials involving over 43,000 women found a statistically significant reduction in breast cancer incidence (HR 0.79; 95% CI, 0.71–0.88), particularly when ET was initiated near the onset of menopause [82].

Data from the Women’s Health Initiative (WHI) estrogen-alone arm further support these findings, reporting a 23% reduction in breast cancer incidence and 44% reduction in mortality after long-term follow-up [83]. Potential mechanisms include downregulation of estrogen receptor expression, reduction in breast epithelial proliferation in the absence of progestins, and systemic metabolic benefits. Despite this, current guidelines restrict ET use to symptomatic menopausal women without a uterus. Nonetheless, these data raise important questions about its role in selected preventive contexts, warranting further dedicated trials with biomarker-enriched risk stratification [68].

### 9.3. Targeting Non-Estrogen Pathway

Given the lack of effective preventive strategies for TNBC and other ER-negative phenotypes, research has increasingly focused on agents that act independently of estrogen signaling. Several repurposed and novel agents are under investigation:

Metformin, an insulin-sensitizing biguanide, has demonstrated antiproliferative and pro-apoptotic effects in preclinical breast cancer models via AMPK activation and mTOR inhibition. Prospective trials such as MA.32 have investigated its role in early-stage breast cancer, and secondary analyses suggest possible risk reduction in insulin-resistant populations [69].

Aspirin, a non-steroidal anti-inflammatory drug (NSAID), has been extensively studied for breast cancer prevention due to its inhibition of COX-2 and downstream effects on inflammation, aromatase activity, and estrogen biosynthesis. Epidemiological studies suggest a modest reduction in breast cancer risk, particularly among postmenopausal women, but randomized trials such as the Women’s Health Study and ASPREE failed to confirm a significant benefit. Given the lack of consistent efficacy and the risk of gastrointestinal and hemorrhagic complications, aspirin is not recommended for primary prevention of breast cancer by current guidelines [69,70,71].

Statins. Observational studies have suggested a potential association between statin use, particularly lipophilic agents such as simvastatin and atorvastatin, and a reduced risk of breast cancer. This effect may be attributed to statins’ inhibition of the mevalonate pathway, leading to reduced cell proliferation and possibly enhanced immune surveillance. However, interventional trials are lacking, and further research is needed to determine whether this association reflects a causal relationship [15].

Retinoids and rexinoids. Retinoids and rexinoids are vitamin A derivatives that act via retinoic acid receptors (RARs) and retinoid X receptors (RXRs) to regulate gene expression involved in cell differentiation, proliferation, and apoptosis. Early retinoids such as all-trans retinoic acid (ATRA) showed anticancer potential but were limited by systemic toxicity. Rexinoids, including bexarotene, selectively target RXRs and have demonstrated greater tolerability and efficacy in preclinical models, notably in reducing ER-negative mammary tumor formation. Phase I trials in high-risk women support their safety profile. Newer agents like UAB30 are under development, offering improved pharmacologic properties. While promising, rexinoids are still under clinical investigation, and larger trials are needed to validate their role in primary prevention [73].

PARP inhibitors. Among breast cancer patients with germline BRCA1/2 mutations, the PARP inhibitors olaparib and talazoparib have demonstrated clinical efficacy in the metastatic setting, as confirmed by the OlympiAD and EMBRACA trials, respectively [74,75,76,77]. More recently, olaparib received approval for adjuvant treatment in patients with HER2-negative early breast cancer and a high risk of recurrence, based on results from the OlympiA trial [76]. Although PARP inhibitors have also been explored in the neoadjuvant setting, their potential role in primary prevention remains undefined [77]. Notably, in the OlympiA trial, nearly half of the participants had undergone bilateral mastectomy, which complicates the evaluation of contralateral breast cancer (CBC) risk. Nevertheless, at a median follow-up of 3.5 years, there was a numerical reduction in both local/regional recurrence and secondary malignancies in patients receiving olaparib compared to placebo. Specifically, fewer cases of invasive CBC and second primary cancers were observed in the olaparib arm [76]. While these findings are encouraging, the preventive impact of PARP inhibitors requires longer follow-up and further dedicated trials. Moreover, their current toxicity profile, particularly hematologic adverse events and the need for close monitoring, poses challenges for their use in healthy high-risk individuals [24,76].

GLP-1 Receptor Agonists. Glucagon-like peptide-1 receptor agonists (GLP-1RAs) are widely prescribed for the treatment of type 2 diabetes and, more recently, for obesity and weight-related metabolic conditions. In addition to improving glycemic control and promoting significant weight loss, growing evidence suggests that GLP-1RAs may also reduce the risk of certain cancers, particularly those associated with obesity, such as breast, endometrial, colorectal, and pancreatic cancers [78,79]. Obesity is a well-established cancer risk factor, driven by chronic low-grade inflammation, insulin resistance, increased estrogen production in adipose tissue, and dysregulated adipokine profiles. These factors create a pro-tumorigenic environment that promotes cell proliferation, angiogenesis, and resistance to apoptosis. By inducing weight loss and improving insulin sensitivity, GLP-1RAs may help reverse many of these pathogenic mechanisms. Moreover, preclinical studies suggest that GLP-1RAs may also exert direct anti-tumor effects, including inhibition of cell growth, induction of apoptosis, suppression of inflammatory pathways, and modulation of the tumor microenvironment [78]. Observational data support this potential. A retrospective cohort study of 1.1 million obese adults (TriNetX database, 2013–2023) used propensity score matching to compare cancer incidence over 5 years in users versus non users of GLP 1 receptor agonists GLP 1RA use was associated with significantly lower risk of several malignancies: gastrointestinal (HR 0.67), skin (HR 0.62), breast (HR 0.72), female genital (HR 0.61), prostate (HR 0.68), and hematopoietic/lymphoid cancers (HR 0.69) [79]. A subsequent systematic review and meta-analysis confirmed a lower incidence of obesity-associated cancers and reduced all-cause mortality among patients with diabetes and obesity treated with GLP-1RAs compared to those using DPP-4 inhibitors [80]. Despite these encouraging findings, important limitations remain. Most available data are observational, with limited cancer-type specificity, especially regarding breast cancer. The absence of randomized trials, uncertainties about optimal duration and timing of treatment, and differences among GLP-1RA agents are unresolved issues. Nevertheless, GLP-1RAs represent a promising class of drugs with potential applications beyond metabolic disease, including cancer prevention. Future prospective studies, particularly in high-risk populations such as postmenopausal women with obesity or metabolic syndrome, will be crucial to validate these early signals and clarify their role in oncologic risk reduction [78,79].

### 9.4. Cancer Vaccines and Immunoprevention

Numerous strategies are under investigation to address the persistent gap in primary immunoprevention for breast cancer, particularly in women at high genetic risk, such as BRCA1 mutation carriers. Non-viral tumor-associated antigens (TAAs) have been characterized that are overexpressed, aberrantly glycosylated, or re-expressed during early tumorigenesis, while remaining largely hidden from immune recognition and non-immunogenic in normal tissues [84]. Among these, MUC1 is one of the most studied. In normal epithelial cells, it is heavily glycosylated and shielded from immune recognition, whereas in tumor cells it is under-glycosylated and overexpressed, becoming immunologically visible and a prime candidate for prophylactic vaccination [85]. HER2 (human epidermal growth factor receptor 2) is another well-characterized antigen, frequently overexpressed in aggressive breast tumors. HER2-targeted vaccines have shown promise in preinvasive settings such as ductal carcinoma in situ (DCIS), inducing antibody and T-cell responses. However, HER2’s low-level expression in some normal tissues raises concerns regarding immune tolerance and safety [86]. In September 2025, the FDA granted Fast Track designation to GLSI-100, a HER2-targeting vaccine (GP2 + GM-CSF) for recurrence prevention in HLA-A02+ HER2+ patients after standard therapy. While this is a secondary-prevention setting, it underscores growing interest in anti-HER2 immunoprevention [87,88]. Mammaglobin A, preferentially expressed in breast tissue and overexpressed in many ER-positive tumors, has been proposed as a subtype-specific vaccine target for hormone-responsive disease [88].

A particularly innovative approach involves “retired self-antigens”—proteins absent in normal adult tissues but re-expressed in tumors. α-Lactalbumin, normally produced only during lactation, is aberrantly expressed in most triple-negative breast cancers (TNBC). Immunologically silent in non-lactating individuals, it offers an opportunity for selective targeting, especially in BRCA1 mutation carriers. Preclinical studies have shown that α-lactalbumin vaccination can induce robust immune responses and prevent TNBC development in murine models, with early-phase clinical trials ongoing [89].

Although non-viral antigens currently dominate breast cancer vaccine research, viral candidates are also being explored. Human mammary tumor viruses (HMTVs), retroviral sequences with >95% homology to the oncogenic mouse mammary tumor virus, are found in up to 38% of breast tumors [90]. HERV-K, an endogenous retrovirus family, expresses envelope proteins on some breast cancer cells and may serve as an additional vaccine target [91]. However, definitive evidence linking these viruses to breast cancer causation remains lacking.

In the pharmacoprevention context, immune checkpoint molecules such as CTLA-4, PD-1, and PD-L1 are emerging targets to restore immune surveillance and intercept tumor development. Blocking these inhibitory pathways—well established in advanced cancer therapy—could, in high-risk individuals, prevent tumorigenesis at premalignant stages [92].

Plasmid DNA vaccines represent another frontier. These synthetic circular DNA constructs encode one or more TAAs and, once delivered intramuscularly with electroporation, transfect antigen-presenting cells to induce endogenous antigen expression and robust CD8^+^/CD4^+^ T-cell activation. Compared with peptide or protein vaccines, plasmid DNA offers greater stability, safety, scalability, and the capacity to induce durable immunity without genomic integration [93]. This approach is particularly suited for genetically predisposed populations, exemplifying a precision interception model that could transform cancer prevention across multiple tumor types.

Collectively, these advances underscore growing interest in prophylactic cancer vaccines targeting tumor-specific or conditionally expressed antigens. Given the heterogeneity of breast cancer, multivalent strategies incorporating several targets may ultimately be required to maximize preventive efficacy.

## 10. Conclusions

Pharmacoprevention is one of the most rigorously evidence-based strategies for reducing the incidence of hormone receptor-positive (HR+) breast cancer in women at increased risk. Despite consistent results from randomized trials and endorsement from international guidelines, clinical uptake remains unacceptably low. This gap reflects a complex interplay of biomedical, behavioral, structural, and sociocultural factors that continue to limit real-world implementation.

Improving uptake will require a multidimensional approach. Incorporating risk-stratified prevention that leverages PRS alongside immune microenvironment features of preinvasive disease may help prioritize candidates, personalize expected benefit, and ultimately improve real-world uptake of pharmacoprevention. Digital tools and AI-driven clinical decision platforms can also help personalize recommendations and reduce decisional conflict. Tailored counseling, simplified dosing strategies such as low-dose or transdermal tamoxifen, and better provider education are essential to enhance adherence and trust. Importantly, pharmacoprevention should not remain confined to high-risk genetic populations or research settings but should become a standard, accessible component of comprehensive cancer prevention pathways.

Looking forward, emerging agents such as oral SERDs, immunoprevention strategies, and metabolically targeted therapies like GLP-1 receptor agonists offer opportunities to expand preventive options beyond endocrine-driven cancers. These innovations exemplify a shift toward precision prevention, where interventions are tailored to individual risk trajectories, genomic profiles, and patient preferences.

Cross-tumor experiences underscore a translational approach: success with aspirin in Lynch syndrome and failures such as β-carotene in smokers highlight that prevention benefits hinge on mechanistic plausibility and context-appropriate selection. These principles should guide next-generation breast cancer pharmacoprevention.

Ultimately, the promise of pharmacoprevention will only be realized through cultural change, system redesign, and integration into national screening and survivorship programs. By transforming prevention into an actionable, personalized strategy, we can empower more women to make informed decisions, reduce cancer incidence, and reshape the future of breast cancer care before the disease begins.

## Figures and Tables

**Table 1 cancers-17-03597-t001:** Absolute risk reduction (ARR), number needed to treat (NNT) and number needed to harm (NNH) in pivotal breast cancer pharmacoprevention trials.

Agent	Trial/Population/mFU	Comparator	Primary Prevention Outcome Used for ARR	ARR	NNT	Key Harm(s) with Absolute Excess → NNH
Tamoxifen 20 mg QD	NSABP P-1 (BCPT) [6]; high-risk women; 5y	Placebo	Invasive breast cancer	2.14%	47	Postmeno. endometrial cancer: +2.3/1000 → NNH 435; PE: +0.7/1000 → NNH 1429; Stroke: +0.9/1000 → NNH 1111; DVT +0.6/1000 → NNH 1667
Tamoxifen 20 mg QD	IBIS-I [14]; high-risk women; 16y	Placebo	All breast cancer	1–2%	—	Harms consistent with SERM class effect (↑ VTE, ↑ endometrium post-meno)
Raloxifene 60 mg QD	STAR (P-2) [15]; postmenopausal high-risk; 6.75y	Tamoxifen	Invasive breast cancer	0	—	Fewer harms vs. tamoxifen; thromboembolism: −1.10/1000 → NNTB 909; invasive endometrial cancer: −0.75/1000 → NNTB 1333; cataracts −2.58/1000 → NNTB 388
Raloxifene 60 mg QD	CORE [8,16]^,^ (extension of MORE [8] trial); osteoporotic postmenopause; 4y-safety data also from RUTH [17] trial; CV risk women; 5.6y	Placebo	Invasive breast cancer	1.19%	84	↑ VTE in RUTH: +1.2/1000 woman-years (NNH 833 per year; 149 over 5.6 y)
Anastrozole 1 mg QD	IBIS-II [10,11] postmenopausal high-risk; 7y	Placebo	All breast cancer	2.8%	36	Fractures not significantly increased (NNH not estimable); BMD loss observed
Exemestane 25 mg QD	MAP.3 [18]; postmenopausal ≥1 risk factor; 3y	Placebo	Invasive breast cancer	1.07%	94	No significant ↑ in major AEs; modest BMD decline
Low-dose Tamoxifen 5 mg QD (3 y)	TAM-01 [19,20] DCIS/ADH/ALH/LCIS after surgery; 5.1y	Placebo	New breast events (invasive + in situ)	4.6%	22	Serious AEs not increased vs. placebo; very low absolute counts → NNH not estimable

Abbreviations: OD, once a day; mFU, median follow-up; CV, cardiovascular; ARR, absolute risk reduction; NNT, number needed to treat; NNH, number needed to harm; NNTB, number needed to treat for benefit when comparing two active agents; Postmeno., postmenopausal; BMD, bone mineral density; PE, pulmonary embolism; VTE, venous thromboembolism; mFU, median follow-up. Notes: (1) ARR/NNT are shown on the trial’s stated median follow-up (mFU), not annualized. (2) For STAR, efficacy is vs. tamoxifen (active comparator); placebo-controlled raloxifene data shown from osteoporosis trials to give an ARR/NNT frame but in a different population. (3) For AIs, fracture risk signals are class-typical; in IBIS-II and MAP.3 the absolute fracture excess was not statistically significant, though BMD loss warrants monitoring. (4) Low-dose exemestane: to our knowledge there is no published phase III randomized prevention trial; omitted here to remain methodologically consistent. (5) ↑, increased risk.

**Table 2 cancers-17-03597-t002:** Comparative features of breast cancer risk stratification models.

Model	Variables	Risk Estimates	Strengths	Limitations	Guideline Thresholds *	Access/Platform
Gail (BCRAT)	Age; reproductive history (menarche, first live birth); first-degree family history; number of breast biopsies; race/ethnicity	5-year and lifetime risk of invasive breast cancer	Simple, fast; widely used; basis of traditional U.S. eligibility criteria	Underestimates risk with strong hereditary history; no genetic variants included	U.S.: ≥1.66% 5-year risk traditionally used for SERM eligibility	Public web calculator (BCRAT)
Tyrer-Cuzick (IBIS)	BMI, HRT use, age at menopause; detailed family history; BRCA1/2 status; mammographic density	10-year and lifetime breast cancer risk	More comprehensive; better identification of hereditary risk; density integration improves discrimination	—	Europe (NICE/ESMO): ≥3% 5-year risk often used in high-risk pathways	IBIS calculator (web/app)
BOADICEA (CanRisk)	Detailed pedigree; pathogenic variants (BRCA1/2, PALB2, CHEK2, ATM); tumor pathology; can integrate other factors	Carrier probability for pathogenic variants; breast & ovarian cancer risks	Genetics-integrated; standard in genetic counseling; precision stratification	—	Used within genetics-informed pathways; thresholds vary by context	CanRisk web

Abbreviations: BCRAT, Breast Cancer Risk Assessment Tool; PRS, Polygenic Risk Score. * U.S. guidelines traditionally use Gail 5-year risk ≥ 1.66% to consider SERM eligibility, whereas European guidelines (e.g., NICE/ESMO) often adopt ≥3% over 5 years to prioritize very high-risk women.

**Table 3 cancers-17-03597-t003:** Guidelines and Recommendations for Endocrine Pharmacoprevention.

Organization	Eligible Population/Risk Criteria	Recommended Agents (Premenopausal/Postmenopausal)	Guideline Thresholds	Recommended Duration	Special Considerations
USPSTF [42]	Women at increased risk and low risk of adverse effects	Tamoxifen/Tamoxifen, Raloxifene, Anastrozole, Exemestane	Gail 5-year risk ≥ 1.66%; other validated models also allowed	5 years	Offer to eligible women; against use in average/low risk. Monitor for VTE/endometrium with SERMs, bone loss with AIs
ASCO [43]	Women at increased risk (models, clinical factors, family history)	Tamoxifen/Tamoxifen, Raloxifene, Anastrozole, Exemestane	Absolute risk: no fixed thresholds recommended; focus on individualized benefit-risk	5 years	Individualize by menopause/comorbidities; support shared decisions
NCCN [43]	Women with high-risk histology (ADH/ALH/LCIS), strong family history, or pathogenic variants (e.g., BRCA1/2)	Tamoxifen/Tamoxifen, Raloxifene, Anastrozole, Exemestane	Gail 5-year risk ≥ 1.7%; other validated models accepted	5 years	Raloxifene if endometrial/VTE risk; Ais accepted but bone monitoring needed
ESMO [33]	High-risk women (clinical/familial); management recommended in specialized clinics.	Tamoxifen/Tamoxifen, Raloxifene, Anastrozole, Exemestane	Tyrer–Cuzick 5-year risk ≥3%	5 years	Recommends individualized discussion and integration into European healthcare pathways
NICE [32]	Women at moderate or high familial risk, preferably assessed in specialized clinics	Tamoxifen/Anastrozole, Tamoxifen, Raloxifene	Tyrer–Cuzick 10-year/lifetime risk 17–29% (moderate) or ≥30% (high)	5 years	Monitor bone health with AIs; integrate with familial/genetic risk management

Abbreviations: USPSTF, United States Preventive Services Task Force; ASCO, American Society of Clinical Oncology; NCCN, National Comprehensive Cancer Network; ESMO, European Society for Medical Oncology; NICE, National Institute for Health and Care Excellence, UK.

**Table 4 cancers-17-03597-t004:** Ongoing Clinical Trials in Endocrine Pharmacoprevention.

Trial/NCT	Design and Population	Intervention/Comparator	Primary/Secondary Endpoints	Status/Sponsor
BABY-TEARS (NCT06364267)	Phase II, randomized, double-blind; postmenopausal women at high risk	Low-dose tamoxifen 10 mg EOD vs. low-dose exemestane 25 mg EOD	Primary: MENQOL; Secondary: Ki-67, mammographic density, adherence, acceptability	Planned start Sept 2025; Academic/Independent
TOLERANT (NCT06033092)	Phase II, 4-arm biomarker study; women at increased risk	Low-dose tamoxifen 10 mg EOD ± lifestyle (ICR, exercise)	Primary: SHBG modulation; Secondary: inflammatory/metabolic markers, microbiome, MD, PROs	Ongoing; Co-funded by the European Commission
LoTam (NCI 2024 06672)	Phase III, randomized; postmenopausal women with low-risk ER+ early BC or intraepithelial lesions	Tamoxifen 5 mg/day vs. standard endocrine therapy (tamoxifen 20 mg or AIs)	Primary: Non-inferiority for recurrence prevention; Secondary: AEs, QoL, adherence, biomarkers	Ongoing; NCI
ELDER	Phase II; postmenopausal women with stage 0–II ER+ BC, pre-surgery	Exemestane 25 mg EOD or 3×/week vs. daily dosing	Primary: Estrogen suppression; Secondary: side effects, adherence	Supported by NCI and BCRF

Abbreviations: NCT, National Clinical Trial identifier; NCI, National Cancer Institute; EOD, every other day; MENQOL, Menopause-Specific Quality of Life questionnaire; SHBG, Sex Hormone Binding Globulin; ICR, Intermittent Caloric Restriction; MD, mammographic density; PROs, patient-reported outcomes; ER, estrogen receptor; BC, breast cancer; AI, Aromatase Inhibitor; BCRF, Breast Cancer Research Foundation.

**Table 5 cancers-17-03597-t005:** Future Directions in Pharmacoprevention.

Strategy/Drug Class	Mechanism of Action	Examples	Key Evidence	Potential Role	Limitations/Challenges
Next-Generation Oral SERDs	Degradation of estrogen receptor; block ER signaling	Elacestrant, Camizestrant, Imlunestrant, Amcenestrant	Phase II–III trials (EMERALD, ELEVATE, EMBER-3) show activity in ER+ BC; under evaluation in prevention	Potential safer, more effective endocrine prevention option	Long-term safety in healthy women unknown; adherence and cost issues
Estrogen-Only Therapy	Hormone replacement without progestins	Conjugated equine estrogens	WHI and meta-analyses: reduce BC incidence in hysterectomized women	Potential preventive option in selected hysterectomized women	Risk of stroke, thromboembolism; limited to hysterectomized population
Non-Estrogenic Targets	Metabolic, anti-inflammatory, DNA repair, and signaling pathways modulation	Metformin, aspirin, statins, rexinoids, PARP-i, GLP-1RAs	Evidence from observational studies, early-phase trials, and preclinical models	Potential complements or alternatives to endocrine prevention, with dual benefits (e.g., metabolic syndrome and cancer risk reduction)	Optimal population selection and safety validation; need for development of new agents in the same class
Immuno-prevention	Induction of tumor-specific immune responses; targeting early oncogenic drivers	HER2-derived peptide vaccines, MUC1 vaccines, neoantigen-based platforms	Early-phase trials show immunogenicity and safety; ongoing studies in high-risk women	Long-term immune-mediated protection in genetically or clinically high-risk populations	Uncertain durability of immune response; challenges in antigen selection, regulatory pathways, and trial design

Emerging pharmacological strategies for breast cancer prevention under clinical or preclinical evaluation. Abbreviations: SERD, Selective Estrogen Receptor Degrader; WHI, Women’s Health Initiative; GLP-1RA, Glucagon-Like Peptide-1 Receptor Agonist; PARP-i, Poly-ADP Ribose Polymerase inhibitor; HER2, human epidermal growth factor receptor 2; MUC1, mucin 1.

## Data Availability

No new data were created or analyzed in this study. Data sharing is not applicable to this article.

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
