# Peer review of "Pharmacological Prevention in Breast Cancer: Current Evidence, Challenges, and Future Directions"

_cancers, 2025, doi:10.3390/cancers17223597_

Round 1
Reviewer 1 Report
Comments and Suggestions for Authors
This is a comprehensive, well-structured, and timely review that thoroughly summarizes the current evidence, challenges, and future directions in breast cancer pharmacoprevention. The authors successfully synthesize a large volume of data from key trials and guidelines, providing a valuable resource for clinicians and researchers. The main weakness is the missed opportunity to use visual aids to summarize complex information, which would significantly enhance the paper's clarity, impact, and reader engagement.
Specific Comments:
(1) The introduction would benefit from being separated into two distinct parts to first establish the critical necessity for primary prevention before introducing the central paradox of low uptake despite strong evidence;
(2) A summary table or figure comparing the efficacy, safety, and indications of the established pharmacological agents (tamoxifen, raloxifene, anastrozole, exemestane) is lacking and is needed for quick clinical reference;
(3) The overview of risk stratification models (Gail, Tyrer-Cuzick, BOADICEA) is descriptive but would be clearer with a comparative table highlighting the unique inputs and strengths of each model;
(4) Table 1 (Guidelines and Recommendations) is incomplete and poorly formatted, making it difficult to compare recommendations across different organizations effectively;
(5) The crucial discussion on Number Needed to Treat (NNT) and Number Needed to Harm (NNH) for various agents and risk groups is presented only in text; a visual representation of this benefit-risk balance would be highly impactful;
(6) The extensive section on future directions (SERDs, GLP-1RAs, vaccines, etc.) is informative but dense; a visual pipeline graphic would greatly improve the presentation of these novel strategies and their developmental stages;
(7) The pivotal results from the TAM-01 trial on low-dose tamoxifen, which is a major focus of the review, are not supported by a visual element (e.g., Kaplan-Meier curve) to illustrate the significant reduction in recurrence rates.
Author Response
Comment Comments and Suggestions for Authors : “The main weakness is the missed opportunity to use visual aids to summarize complex information, which would significantly enhance the paper's clarity, impact, and reader engagement.”.
Response Comments and Suggestions for Authors: We sincerely thank the Reviewer for this constructive suggestion. We would like to clarify that a Graphical Abstract was prepared and submitted with the original version of the manuscript (uploaded in the Supplementary Material), which we believe effectively summarizes the main concepts and flow of the paper. We regret if this was not easily accessible during the review process. In addition, in the revised version of our manuscript we have further strengthened the use of visual aids by including Table 1 (summarizing absolute risk reduction, number needed to treat, and number needed to harm for established agents) and have updated Table 3 (summarizing ongoing trials) and Table 4 (Future Directions in Pharmacoprevention). We believe that the combination of the Graphical Abstract and these visual summaries significantly improve clarity, impact, and reader engagement, fully addressing the Reviewer’s concern .
We invite the Reviewer to refer to the Graphical Abstract and to Tables 1, 3, and 4 for a concise visual overview of these aspects.
Comments 1 “The introduction would benefit from being separated into two distinct parts to first establish the critical necessity for primary prevention before introducing the central paradox of low uptake despite strong evidence;”.
Response 1 We thank the Reviewer for this thoughtful suggestion. We fully agree that the Introduction should first emphasize the necessity of primary prevention in modern oncology, and then introduce the issue of low uptake despite strong supporting evidence. Our original text already followed this logical structure; however, in the revised version we have refined it further by explicitly separating the two parts and adding clearer transitions. In particular, the Introduction now opens with a broader discussion of the importance of prevention and subsequently introduces a dedicated subsection entitled “From Evidence to Practice: The Uptake Gap”. We believe that these modifications make the narrative more transparent and aligned with the Reviewer’s recommendation (page 2, lines 30–75 of the revised manuscript).
Comment 2 A summary table or figure comparing the efficacy, safety, and indications of the established pharmacological agents (tamoxifen, raloxifene, anastrozole, exemestane) is lacking and is needed for quick clinical reference;
Response 2 We thank the Reviewer for this very helpful suggestion. In the revised version, we have included a new Table 1 that summarizes the pivotal prevention trials with tamoxifen, raloxifene, anastrozole, and exemestane. For each agent, the table reports the trial population, median follow-up, absolute risk reduction, number needed to treat, and number needed to harm, together with the most relevant safety signals. We believe this visual summary provides a concise and clinically useful reference that facilitates comparison across agents and significantly improves the clarity and impact of the manuscript (Table 1; page 2, about lines 64 of the revised version).
Comment 3 The overview of risk stratification models (Gail, Tyrer-Cuzick, BOADICEA) is descriptive but would be clearer with a comparative table highlighting the unique inputs and strengths of each model;
Response 3 We thank the Reviewer for this helpful suggestion. In the revised version, we have added a new Table 2 that provides a comparative summary of the Gail (BCRAT), Tyrer–Cuzick (IBIS), and BOADICEA (CanRisk) models. The table highlights the key risk factors included, the type of risk estimates provided, their unique strengths, limitations, commonly applied guideline thresholds, and platforms for access. In addition, we have included a clarifying note regarding the different thresholds traditionally used in U.S. (≥1.66% 5-year risk with the Gail model) versus European guidelines (≥3% 5-year risk, e.g., NICE/ESMO) for pharmacoprevention eligibility. We believe this addition improves clarity, provides a quick clinical reference, and addresses the Reviewer’s concern.
We invite the Reviewer to refer to Table 2 for a concise visual overview of these aspects
Comment 4 Table 1 (Guidelines and Recommendations) is incomplete and poorly formatted, making it difficult to compare recommendations across different organizations effectively;
Response 4 We thank the Reviewer for this valuable comment. In the revised manuscript, we have thoroughly restructured Table 1 (now renamed Table 3 in the revised version) to provide a clear and comparative summary of the major international guidelines (USPSTF, ASCO, NCCN, ESMO, NICE). The table has been reformatted with standardized columns, including eligible population/risk criteria, recommended agents (premenopausal vs postmenopausal), guideline thresholds, recommended duration, and key notes/special considerations. Spelling and formatting errors were corrected, and information on thresholds, duration of treatment, and clinical considerations (e.g., bone health monitoring, familial/genetic risk integration) has been harmonized across organizations. We believe that the revised Table 3 (ex Table 1) now offers a concise, accurate, and easily comparable overview, directly addressing the Reviewer’s concern.
We invite the Reviewer to refer to Table 3 for a concise visual overview of these aspects
Comment 5 The crucial discussion on Number Needed to Treat (NNT) and Number Needed to Harm (NNH) for various agents and risk groups is presented only in text; a visual representation of this benefit-risk balance would be highly impactful;
Response 5 We thank the Reviewer for this important suggestion. In the revised version, we have added a new Table 1 summarizing the pivotal phase III prevention trials. The table reports absolute risk reduction (ARR), Number Needed to Treat (NNT), and Number Needed to Harm (NNH) for tamoxifen, raloxifene, anastrozole, exemestane, and low-dose tamoxifen. This visual summary complements the text, provides an immediate benefit–risk comparison across agents, and offers a practical clinical reference. We believe this addition directly addresses the Reviewer’s concern and improves the clarity and impact of the manuscript.
We invite the Reviewer to refer to Table 1
Comment 6 The extensive section on future directions (SERDs, GLP-1RAs, vaccines, etc.) is informative but dense; a visual pipeline graphic would greatly improve the presentation of these novel strategies and their developmental stages;
Response 6 We thank the Reviewer for this insightful suggestion. In the revised manuscript, we have substantially improved Table 3 (now renamed Table 5, in the revised version Future Directions in Pharmacoprevention) by restructuring it into a more concise and comparative format, which we believe significantly enhances clarity and readability. In addition, we would like to highlight that the manuscript is already accompanied by a Graphical Abstract, which provides a visual synthesis of emerging strategies and their developmental context. We hope that these improvements satisfactorily address the Reviewer’s concern.
We invite the Reviewer to refer to Table 3 for a concise visual overview of these aspects
Comment 7 The pivotal results from the TAM-01 trial on low-dose tamoxifen, which is a major focus of the review, are not supported by a visual element (e.g., Kaplan-Meier curve) to illustrate the significant reduction in recurrence rates.
Response 7 We thank the Reviewer for this thoughtful observation. While we fully agree on the importance of the TAM-01 trial in shaping the field, we do not consider it to be the exclusive focus of our review, which aims to provide a broader overview of both established and emerging strategies in breast cancer pharmacoprevention. For this reason, we chose not to reproduce Kaplan–Meier curves already available in the original publication. Instead, to enhance clarity and clinical relevance, we have created a new table (Table 1), which systematically summarizes the pivotal prevention trials, including TAM-01, and reports key measures such as absolute risk reduction (ARR), number needed to treat (NNT), and number needed to harm (NNH). We believe this provides a concise and practical representation of the trial’s impact, fully aligned with the comparative scope of our review.
We invite the Reviewer to refer to Table 1 for a concise visual overview of these aspects
Reviewer 2 Report
Comments and Suggestions for Authors
The authors have pulled together a large amount of data on SERMs, aromatase inhibitors, and newer approaches such as low-dose tamoxifen and digital tools for adherence. The subject is clearly relevant, and I appreciate that the paper is framed around both established evidence and future possibilities. That said, in its current form the paper feels more like a catalog of information than a cohesive review, and it really needs re-structuring and deeper contextualization before it can be considered for publication.
My main impression is that the introduction is too narrow and jumps straight into pharmacoprevention without adequately setting the stage for why this matters in the broader landscape of cancer treatment and prevention. Since this is a cancer study, I would strongly recommend beginning with a more cohesive overview of cancer biology and current treatment strategies, which would give general readers a stronger clinical anchor before diving into pharmacological prevention. For example, the review by D. Sonkin and A. Thomas, “Cancer Treatments: Past, Present, and Future” (2024), authored by the Chief of the US National Cancer Institute, is a very useful reference to situate pharmacoprevention within the wider arc of oncology progress. Similarly, citing broader conceptual works like “Different strategies for cancer treatment: targeting cancer cells or their neighbors?” ( 2025) would help connect breast cancer prevention to the ongoing debate over whether we target tumors directly or modulate their environment.
Another area that needs strengthening is the discussion of statistics and quantitative framing. The paper does cite many trials, but the way the numbers are presented is inconsistent. Sometimes relative risk reductions are highlighted, other times absolute numbers are missing. To make this a stronger and more reader-friendly review, the authors should systematically present absolute risk reductions, number needed to treat, and number needed to harm, ideally in tables or side-by-side comparisons. That would make the case for pharmacoprevention far clearer. Right now, the statistical evidence feels scattered and hard to digest.
The structure is also an issue. Sections on SERMs and AIs are comprehensive, but later parts (digital health tools, low-dose tamoxifen, ongoing trials) read more like appendices than integrated arguments. The authors should re-organize the narrative so that the story builds logically: first, why prevention is important in the context of modern oncology; second, what the strongest current evidence is (SERMs and AIs); third, why uptake remains low; and finally, what emerging strategies are showing promise. Without that, the paper risks overwhelming the reader with detail while underselling its own significance.
I would also encourage the authors to bring in more mechanistic or translational insights, not just trial outcomes. For example, how do genomic risk scores or immune-related markers intersect with the uptake of pharmacoprevention? Are there parallels with other cancers where prevention has succeeded or failed? This would enrich the discussion and broaden its relevance.
Author Response
Comments 1: in its current form the paper feels more like a catalog of information than a cohesive review, and it really needs re-structuring and deeper contextualization before it can be considered for publication
Response 1: In accordance with the reviewer’s comment, we have carefully reviewed the paper and hope that our revisions address all the concerns raised. Please find below our detailed, point by point rebuttal.
Response 2: We thank the reviewer for this valuable suggestion. In accordance with the comment, we have revised and expanded the Introduction to provide a broader context for cancer biology and treatment, also citing the recommended works by Sonkin et al. (2024) and Liu et al. (2025) (page 1,2, lines: 41-47 page 10, lines 476-483).
Comment 3: “I would also encourage the authors to bring in more mechanistic or translational insights, not just trial outcomes. For example, how do genomic risk scores or immune-related markers intersect with the uptake of pharmacoprevention? Are there parallels with other cancers where prevention has succeeded or failed? This would enrich the discussion and broaden its relevance.”.
Response 3: We thank the reviewer for this insightful comment. In response, we have expanded the sections “Future Directions in Pharmacoprevention” and “Conclusions” to incorporate mechanistic and translational perspectives, including the role of genomic risk scores, immune-related markers, and cross-tumor lessons from successful and unsuccessful prevention strategies. (page 13-16 lines: 624-828; page 17, lines: 832-861).
Response 4: We thank the reviewer for this important suggestion. In response, we have added a new table (new Table 1) reporting absolute risk reductions, numbers needed to treat, and numbers needed to harm from the major phase III prevention trials, thereby providing a standardized quantitative comparison across agents. We have also explicitly referred to this table in the revised text (page 2, lines: 80-87. page 11, lines 567,569). We invite the Reviewer to refer to Table 1 for a concise visual overview of these aspects
Comment 5: "The authors should re-organize the narrative so that the story builds logically: first, why prevention is important in the context of modern oncology; second, what the strongest current evidence is (SERMs and AIs); third, why uptake remains low; and finally, what emerging strategies are showing
Response 5: we thank the Reviewer for this important observation. In the revised version, we have substantially restructured the sections on digital health tools, low-dose tamoxifen, and ongoing trials. Specifically, we eliminated the previous sub-paragraph format and integrated the content into cohesive narrative sections, ensuring smoother transitions and logical connections with the earlier discussion on SERMs and AIs. We also added explicit cross-references to new Table 4 ( previous table 2) and to related sections (e.g., linking low-dose tamoxifen to ongoing trials), to reinforce their role as integral components of the overall argument rather than appendices. We believe these changes have improved the structural coherence and readability of the manuscript (page 12, lines: 618-619 pages 14-16 lines 648-828).We invite the Reviewer to refer to Table 4 for a concise visual overview of these aspects
Comment 6: "The authors should re-organize the narrative so that the story builds logically: first, why prevention is important in the context of modern oncology; second, what the strongest current evidence is (SERMs and AIs); third, why uptake remains low; and finally, what emerging strategies are showing promise. Without that, the paper risks overwhelming the reader with detail while underselling its own significance."
Resppnse 6 We sincerely thank the Reviewer for this valuable suggestion, which we fully agree has improved the readability and overall narrative of the manuscript. In the revised version, we have adjusted the titles of the major sections to explicitly reflect the logical progression proposed—beginning with the importance of prevention in modern oncology, followed by the strongest current evidence (SERMs and AIs), the reasons for limited uptake, and finally the emerging strategies that hold promise. While the original structure already followed this progression in substance, we recognize that making it more explicit through section headings strengthens the clarity and accessibility of the review. We believe that these changes better highlight the significance of the field and enhance the coherence of our presentation.
Reviewer 3 Report
Comments and Suggestions for Authors
1). Line 177: 'Populations' should be written with lower case 's'.
2). Table 1: numerous spelling mistakes on the first row (title row).
3). Line 336: for citation 3, what is the follow-up saying? That is an old paper, so what progress has been made to date since?
4). Line 399: giving drugs to those who do not need it is not ethically justifiable. The progression to disease from a 'non-existent/preventable' state can never be assessed, so this will always be a tricky situation for a physician to bring up with the patient. That plus the fact that drugs will cost money and have undoubtedly some form of side effects lead the reasons for resistance/non-compliance.
5). Line 435: 'intolerability' suggests that drugs are not that safe/compatible. This ties in with my concerns above.
6). Line 450: it is concerning to this reviewer that this review is very pro-pharmacoprevention, which may be balanced (as some form of remedy) by providing readers the dangers of administration of drugs that could have side effects. I am also concerned that such lobbying just falls within the self-serving wishes of pharmaceutical companies, which could be a slippery slope. Perhaps, a good (ethical) around this dilemma is to provide a more balanced (non-one-sided) view of the area, regardless of where the authors allegiances lie.
7). Line 623: at low dose chemotherapy, would there not be a risk of development of drug resistance? Comment using appropriate citations to present pros and cons.
8). Line 809: citation [96] has nothing to do with showing positive attributes of pDNA use as vaccines.
Author Response
Comment 1: Line 177: 'Populations' should be written with lower case 's'
Response1: We thank the Reviewer for this observation. The word has been corrected in the revised manuscript.
Comment 2 Table 1: numerous spelling mistakes on the first row (title row).
Response2: We thank the Reviewer for pointing this out. In the revised manuscript, the former Table 1 has been completely reviewed and reformatted, with all spelling mistakes corrected and the structure harmonized across columns to ensure clarity and consistency. This table is now numbered as Table 3 in the revised version.
Comment 3: Line 336: for citation 3, what is the follow-up saying? That is an old paper, so what progress has been made to date since?
Response 3: We thank the Reviewer for this important observation. Citation [3] originally referred to the initial report of the STAR trial (JAMA 2006). In the revised manuscript, we now clarify that these conclusions are supported not only by the initial results but also by the extended follow-up analyses (Martino et al., Cancer Prev Res 2010), which confirmed the trial’s findings. We have accordingly updated the references in the text and bibliography.
Comment 4: Line 399: giving drugs to those who do not need it is not ethically justifiable. The progression to disease from a 'non-existent/preventable' state can never be assessed, so this will always be a tricky situation for a physician to bring up with the patient. That plus the fact that drugs will cost money and have undoubtedly some form of side effects lead the reasons for resistance/non-compliance.
Response 4: We appreciate the Reviewer’s perspective. The ethical concerns regarding the use of preventive pharmacological agents in otherwise healthy women, such as cost, potential side effects, and the uncertainty of whether an individual would ever progress to disease, are extensively addressed in Sections 5 and 6 of the manuscript. To further enhance clarity, we now present in new Table 1 the absolute risk reduction (ARR), number needed to treat (NNT), and number needed to harm (NNH) from the pivotal pharmacoprevention trials, which provide a transparent quantitative framework for weighing efficacy against toxicity. We would also like to emphasize that the current challenge is not overuse of pharmacoprevention, but rather its strikingly low uptake despite strong evidence of benefit in appropriately selected high-risk women, which is one of the central issues this review aims to address.
We invite the Reviewer to refer to Table 1 for a concise visual overview of these aspects
Comment 5: Line 435: 'intolerability' suggests that drugs are not that safe/compatible. This ties in with my concerns above
Response 5: We thank the Reviewer for this comment. We agree that intolerability is an important and well-documented issue in clinical practice, contributing substantially to the high discontinuation rates observed with preventive therapies. In our review, we considered this aspect as part of a broader discussion on the barriers that explain the historically low uptake of pharmacoprevention. Importantly, we also outline strategies that may mitigate these challenges, such as better patient selection through refined risk models, improved communication of the risk-benefit balance, digital tools to support adherence, shorter or low-dose regimens, and newer agents with more favorable safety profiles. Our aim is to acknowledge intolerability as a real-world limitation while framing it within the wider context of current barriers and potential solutions for more effective and acceptable pharmacoprevention.
Comment 6: Line 450: it is concerning to this reviewer that this review is very pro-pharmacoprevention, which may be balanced (as some form of remedy) by providing readers the dangers of administration of drugs that could have side effects. I am also concerned that such lobbying just falls within the self-serving wishes of pharmaceutical companies, which could be a slippery slope. Perhaps, a good (ethical) around this dilemma is to provide a more balanced (non-one-sided) view of the area, regardless of where the authors allegiances lie.
Response 6: We thank the Reviewer for this thoughtful comment. We respectfully confirm that our review is intentionally pro-pharmacoprevention, because it reflects the evidence that preventive therapy remains strikingly underused despite strong data from pivotal clinical trials. This underutilization represents a missed opportunity to reduce breast cancer incidence, which continues to show a gradual increase in many Western countries (estimated annual rise of 0.5-1% in age-standardized incidence rates). We believe that a transparent presentation of both benefits and limitations is essential. In this respect, our revised manuscript now provides a more quantitative appraisal of benefit-risk trade-offs through Table 1, which reports absolute risk reduction (ARR), number needed to treat (NNT), and number needed to harm (NNH) from pivotal trials. Thus, potential side effects and adherence barriers are not minimized but rather explicitly integrated into our analysis. We would also like to emphasize that all pharmacological agents currently recommended for prevention (tamoxifen, raloxifene, anastrozole, exemestane) are generic, off-patent drugs with low cost, and therefore not subject to promotion by pharmaceutical companies. Indeed, the lack of commercial interest may partly explain the absence of public campaigns and the persistent low adoption of pharmacoprevention in clinical practice. Our conviction remains that prevention is better than cure, not only in terms of public health outcomes but also in reducing the enormous economic burden of treating advanced breast cancer. Finally, we view the Reviewer’s point as a valuable suggestion for the Editor: this review could be complemented within a Pro and Contra framework, by inviting additional contributions with alternative or critical perspectives, thus enriching the debate around pharmacoprevention.
Comment 7: Line 623: at low dose chemotherapy, would there not be a risk of development of drug resistance? Comment using appropriate citations to present pros and cons.
Response 7: We thank the Reviewer for raising this point. We would like to clarify that the agents under evaluation for low-dose strategies, tamoxifen and exemestane, are endocrine therapies, not cytotoxic chemotherapies. The concern that suboptimal dosing may foster drug resistance is well established for classical chemotherapy, where dose intensity directly correlates with clonal selection and resistance (Norton & Simon, Cancer Treat Rep 1977). In contrast, endocrine prevention trials such as TAM-01 (low-dose tamoxifen) and ELDER (reduced-dose exemestane) are specifically designed to identify the minimum effective dose and duration, with the goal of maintaining efficacy while improving safety and adherence in otherwise healthy women. To address this more explicitly, we have modified the text in the section Emerging Strategies: Ongoing Clinical Trials as follows: “Unlike cytotoxic chemotherapy, where suboptimal dosing may promote resistance [new ref. 55], all these trials are designed to identify the minimum effective dose and duration, balancing efficacy with tolerability. They illustrate a paradigm shift toward precision prevention, emphasizing patient-centered, low-toxicity, and biomarker-guided strategies.” We believe this clarification directly addresses the Reviewer’s concern while preserving the focus of our manuscript.
(page 12, lines: 612-619). Emerging Strategies: Ongoing Clinical Trial
Comment 8: Line 809: citation [96] has nothing to do with showing positive attributes of pDNA use as vaccines.
Response 8: We sincerely thank the Reviewer for pointing this out. We apologize for the oversight. In the revised version, the entire bibliography has been carefully reviewed, corrected, and renumbered, ensuring that all references accurately correspond to the citations in the text. [91]
Round 2
Reviewer 1 Report
Comments and Suggestions for Authors
Authors have addressed all of my concerns, it could be accepted for publication now.
Reviewer 2 Report
Comments and Suggestions for Authors
good
Comments on the Quality of English Languagegood